# Origin and Enrichment Mechanisms of Salinity and Fluoride in Sedimentary Aquifers of Datong Basin, Northern China

**DOI:** 10.3390/ijerph20031832

**Published:** 2023-01-19

**Authors:** Xianguo Wang, Ranpatiyalage Nishamani Nuwandika Weerasinghe, Chunli Su, Mengzhu Wang, Jiaqi Jiang

**Affiliations:** 1Henan Geological Engineering Survey Institute, Zhengzhou 450001, China; 2Department of Science, National Institute of Education, Maharagama 10280, Sri Lanka; 3School of Environmental Studies, China University of Geosciences, Wuhan 430078, China; 4State Environmental Protection Key Laboratory of Source Apportionment and Control of Aquatic Pollution, China University of Geosciences, Wuhan 430078, China

**Keywords:** enrichment mechanism, salinity, hydrogeochemistry, influencing factors, statistical analyses

## Abstract

The exposure of inhabitants to high fluoride and saline groundwater is the main health issue in Datong Basin, Northern China. This study aims to elucidate the spatial distribution and the mechanisms of high fluoride and salinity occurrence in the shallow sedimentary aquifers of the Datong Basin. Groundwater salinity and fluoride content, and their association with measured hydrochemical parameters, were conducted using multivariate statistical analyses. The analytical results revealed that the concentrations of fluoride and total dissolved solids (TDS) show dramatic variations within the study area. Around 41.4% of groundwater samples contained high-level fluoride concentration (F^−^ > 1.5 mg/L), whereas 32.8% contained elevated-level TDS (TDS > 1000 mg/L). Both fluoride and TDS concentrations had elevated trends towards the central part of the basin. Shallow groundwater was seriously affected by evaporation and evapotranspiration, which can be the critical factors responsible for rather high TDS and F^−^ concentrations in shallow aquifers. Water–rock reactions including silicate hydrolysis, dissolution–precipitation of carbonates and evaporates, adsorption, and ion exchange processes, as well as evapotranspiration, are the main governing factors for salinity and fluoride enrichment in groundwater. Solubility control of F-bearing and carbonate minerals is the dominant mechanism affecting F^−^ levels. Prevailing conditions of alkaline pH, moderate TDS and Na^+^, high HCO_3_^−^, and lower Ca^2+^ content facilitate the enrichment of fluoride in the study area. Excessive evapotranspiration can be also the most influencing factor responsible for high fluoride and TDS content, due to the extended residence time of groundwater and the arid climate of the central part of the Datong Basin.

## 1. Introduction

Groundwater is the major source of drinking, irrigation, and industrial requirements in arid and semi-arid area, where availability of groundwater and its quality has always been a major concern for local residents [1,2,3]. Rapid population growth and intensified agricultural and industrial demands are driving to the over-exploitation of groundwater. The groundwater has deteriorated in many areas due to contaminants of natural and anthropogenic interventions [4,5]. Salinity in groundwater is one of the prominent phenomena of water quality deterioration, which prevailing often in arid and semi-arid regions. Over the last centuries, a large number of aquifers and some river basins became undesirable for human consumption due to elevated salinity [6]. Natural saline water in arid and semi-arid regions was mainly attributed to the long-term accumulation of salt materials on the earth’s surface and the deficit of ample flushing in the unsaturated zone. The fresh and saline water aquifers are naturally separated by impermeable layers. The artificial combinations created among those impermeable layers due to improper drilling and well construction activities, cause the mixing of different water types. Excessive amounts of inorganic pollutants such as fluoride, sulfate, boron, and bio-accumulated elements (arsenic and selenium) are considered poisonous components in groundwater [7]. Similarly, high saline groundwater occurs when an excessive number of inorganic ions are available in the water.

Fluoride is one of the typical chemical parameters that had a great impact on groundwater quality [8]. Fluoride can be formed by mineral complexes with various cations. Some rare mineral species of low solubility contain fluoride [9]. Although high fluoride concentrations mostly occur in igneous and metamorphic aquifers, it extends to sedimentary aquifers, particularly in arid and semi-arid regions [10]. Fluorine-bearing clays and fluorapatite are common in sedimentary formations because of phosphate-rich water precipitation processes [7]. High fluoride concentrations are naturally occurred in groundwater based on the hydrogeological and geochemical characteristics, buried depth of the aquifers, hydrochemical characteristics, underlying temperature, and the effect of weathering [11,12]. Health problems related to fluoride are common in arid and semi-arid areas of many countries in the world [13,14]. Elevated fluoride concentrations in drinking water (>1.5 mg/L) cause dental and skeletal fluorosis threatening human health. Approximately 200 million people worldwide consume high-fluoride concentrated water [15]. Some examples of high fluoride-concentrated groundwater countries include Canada (15.1 mg/L), Ethiopia (0.01 to 13 mg/L), Ghana (282.29 mg/L), Iran (0.2 to 9.2 mg/L), Korea (40.8 mg/L), Pakistan (0.11 to 22.8 mg/L), Sri Lanka (5 mg/L) and Tanzania (15 to 63 mg/L) [16].

In some regions of China, Africa, and India, elevated fluoride concentrations exist along with saline groundwater and caused severe dental and skeletal fluorosis [17,18]. Chemical weathering occurs in arid and semi-arid areas that result in salinity enhancement as well as elevated fluoride concentrations in phreatic water systems [19]. Although the sources of fluoride and salinity in groundwater have been identified in most regions of the world, it is necessary to further investigate their enrichment mechanisms involving complex processes that control its migration from minerals containing aquifer matrix into groundwater.

High levels of saline and fluoride groundwater have become a serious environmental and health issue in Northern China [19,20,21]. High fluoride groundwater that exceeds the WHO drinking water permissible value of 1.5 mg/L was found widely in the arid and semi-arid areas of about 2,200,000 km^2^ in Northern China [10,22,23,24]. Datong Basin, which is rich with complex sedimentary hydrogeological features, has large ranges of sub-surface solute concentrations. Those solutes are mobilized and stored in the sub-surface leading to increased salinity in groundwater. The exposure of inhabitants to high fluoride and high saline groundwater across the basin is the main health issue at Datong due to inefficient local management to identify alternative sources of water supply. According to official statistics in 2005, 50.85 million people in rural areas of Northern China consumed high-fluoride water [25]. Therefore, it is important to identify the mechanisms of elevated concentrations of salinity and fluoride in the groundwater at the Datong Basin.

In the recent years, extensive studies have been carried out on the distribution, hydrochemical characteristics of high-TDS and high-F^−^ groundwater, health risk assessment and prediction using computer models and mathematical statistics [26,27,28,29,30,31,32]. However, better understanding of the geospatial variations of hydrogeochemical components and associated processes determining groundwater quality is essential for achieving and sustaining usable water supplies. This study aims to illustrate the hydrogeochemistry, spatial occurrence, and a synthesis of mechanisms leading to high-F^−^ and high-TDS in the central parts of Datong Basin. The hydrochemical evolution processes and key factors responsible for high TDS and fluoride level were addressed by combining hydrochemical components, stable hydrogen and oxygen isotopes and principal component analysis. It is anticipated that this study helps in planning the sustainable use of the groundwater in the Datong Basin, Northern China.

## 2. Study Area

Datong Basin is located in the Northern Shanxi Province, China. It covers approximately 6000 km^2^ of the area. The altitude varies between 863 and 1266 m above sea level from Northeast to Southwest. It has a semi-arid climate with an average precipitation of 225–400 mm and potential evaporation above 2000 mm.

This basin is one of the Cenozoic faulted basins of the Shanxi rift system that is surrounded by mountains in the Southeast and Northwest directions [33]. The outcroppings are Archeangneiss and basalt in the north, Cambrian–Ordovician limestones and Carboniferous–Permian–Jurassic sandstones and shale in the west and Archean gneiss and granite sparsely in the northeast. Alluvial fans and alluvial inclined plains are widely distributed in the edge of the basin, and the center of the basin is a slightly inclined broad alluvial lacustrine plain [21]. The thickness of the Quaternary sediments increases from 200 m at the margin to 2700 m in the central part of the basin, and the grain sizes of the sediments generally decrease. The sediments in the central parts of the basin consist of mainly sandy loam and silt, lacustrine and alluvial–lacustrine sandy loam, silty clay and clay with high contents of organic matters, whereas those at the basin margin are mostly alluvial–pluvial gravel and sand [29]. Ephemeral rivers, such as the Sanggan River and Huangshui River, flow though the basin.

The groundwater in Datong Basin mainly occurs in four types of aquifers: phreatic aquifers, shallow semi-confined aquifers, medium confined aquifers with early and middle Pleistocene sand and sandy gravel, and deep confined aquifers with early Pleistocene and Pliocene fine sand and silt [34,35]. The general groundwater flow regimes are the flow from marginal mountain fronts to the center of the basin, and the flow inside the basin along the direction of the river from Southwest to Northeast [34,35]. Groundwater is mainly recharged from vertical infiltration of meteoric water in the basin and the fracture water from bedrock flows laterally along the basin margins. Evapotranspiration and artificial abstraction are the major discharge processes of groundwater in the basin.

## 3. Methodology

### 3.1. Sampling and Analysis

In total, 128 groundwater samples were collected from the drinking wells and irrigation wells within the study area in August 2014 (Figure 1). According to the burial depth and confined pressure of the aquifer, the groundwater samples (*n* = 128) obtained below 50 m and above 50 m were categorized into two distinct groups as shallow (*n* = 81) and deep groundwater samples (*n* = 47), respectively.

The measured parameters include physical parameters (pH, temperature, electrical conductivity (EC), major ions (F^−^, Cl^−^, NO_3_^−^, SO_4_^2−^, HCO_3_^−^, K^+^, Na^+^, Ca^2+^, and Mg^2+^) and some metal elements (Li, Ba, Sr, Fe, Mn, and Pb) as well as stable isotopes of δD and δ^18^O. Temperature, pH, and EC were taken as on-site measurements using a potable Hanna EC and pH meter which was calibrated daily before use. Alkalinity was measured from Gran titration using 0.05 N HCl on the sampling day. Major anions were determined using ion chromatography (DX-120), major cations by ICP-AES (IRIS INTRE II XSP type), and trace metal elements by ICP-MS with a detection limit of 1 μg/L for most elements. Stable isotopes of δD and δ^18^O were measured using MAT251.

### 3.2. Data Processing

All the average values of the measured hydrochemical parameters were compared with the updated local and international guideline standards for different chemical compositions in drinking water. SPSS 19.0 was used for multivariate statistical analysis of hydrochemical data. Principal component analysis (PCA) was employed to identify the underlying hydrogeochemical factors related to fluoride and salinity enrichment of groundwater in the study area. ArcGIS 10.3 was applied to create the distribution maps of simpling sites as well as fluoride and TDS concentrations. The SI of minerals and geochemical inverse modelling were calculated using the hydrochemistry software PHREEQC 3.4.

## 4. Results and Discussion

### 4.1. Hydrochemical Characteristic and Differentiation

The statistics of measured parameters suggest a large spatial heterogeneity of groundwater chemistry in the study area. All the measured hydrochemical parameters of groundwater have been compared with the latest groundwater quality standards (China’s standard for groundwater quality-GB/T 14848-2017 and WHO guideline values for drinking water quality of 2017) (Table 1).

The groundwater pH ranges from 7.0 to 8.7 indicating weak alkaline conditions prevailing in groundwater in the study area. The EC and TDS range from 206 to 22,600 µs/cm and 208.9 to 10,661 mg/L, respectively, reflecting the different amounts of ionic substances in groundwater that may have an influence on salinity in groundwater. The maximum values of EC and TDS have been recorded in shallow groundwater (Table 1). However, EC shows a high variance in deep groundwater demonstrating the existence of overflow recharge from shallow groundwater. Furthermore, 33% (*n* = 42) of the samples exceed the permissible TDS concentration of 1000 mg/L [37,38]. The large variations of TDS may be found due to natural geochemical processes and anthropogenic interventions. 

The dominance of anions is in the order of HCO_3_^−^ > Cl^−^ > SO_4_^2−^ > NO_3_^−^ > F^−^, whereas the concentration of cations is in the order of Na^+^ > Mg^2+^ > Ca^2+^ > K^+^ in the study area. The concentrations of major cations Na^+^, Mg^2+^, and Ca^2+^ have exceeded 2895, 327, and 773 mg/L, respectively, highlighting the maximum recorded values at shallow depths. The concentrations of major anions HCO_3_^−^, SO_4_^2−^, NO_3_^−^, Cl^−^ and F^−^ reached up to 1537, 4456, 1118, 3272, and 8.9 mg/L, respectively. The ionic composition of groundwater shows high variability, and it is confirmed by standard deviations, which are larger than their mean values. Approximately 40% (*n* = 51) of samples contain sodium that exceeds the local permissible value of 200 mg/L with a maximum value of 2895 mg/L, which may alter the taste of water. The majority of shallow groundwater (58%), as well as deep groundwater (80.8%), contain HCO_3_^−^ as the dominant anion. Moreover, 37% (*n* = 47) and 17% (*n* = 22) of samples exceeded the permissible value of nitrates 20 mg/L [36] and 50 mg/L [39]. The high concentrations of nitrates in groundwater may occur owing to intensive agricultural activities and using excessive nitrates-containing fertilizers and manure in some areas of the basin. The sulfate concentration ranges from 0.05 to 4456 mg/L, and contains 25% (*n* = 32) of samples that exceeded the recommended guideline value of 250 mg/L [37,38]. In total, 27% of samples (*n* = 35) have elevated chloride concentration, exceeding the recommended value of 250 mg/L [37,38]. The fluoride concentration ranges from 0.01 to 8.69 mg/L, and 59% (*n* = 75) and 41% (*n* = 53) of samples have higher concentrations compared to the guideline values of 1.0 mg/L [37] and 1.5 mg/L [39], respectively. The metal constituents also exist in low concentrations at Datong Basin. Strontium is the most abundant trace element of the groundwater in this area. Barium, Fe, Mn, and Pb concentrations were within the recommended permissible level (Table 1).

TDS concentration ranges from 208.9 to 10,661 mg/L (SD ± 2021) with a mean value of 1462 mg/L, showing the dramatic variation at Datong Basin (Table 1). The majority (76 samples) of groundwater contained TDS > 500 mg/L conflates with carbonate weathering or sea-water mixing [40]. Out of 128 samples, 42 samples (32.8%) belong to the category of high TDS. Approximately 44.4% (36 samples) of shallow groundwater and 12.8% (6 samples) of deep groundwater contain high TDS concentrations. Therefore, elevated TDS may be a major concern in shallow groundwater than that deep groundwater. The majority of high TDS shallow groundwater contains HCO_3_-Na-type water with temporary hardness. This may suggest that the evaporation process which is severely affected by shallow groundwater compared to deep groundwater, can be a dominant process of the salinity enrichment. Salinity can be interpreted using EC values. EC is directly related to dissolved ionic solutes of the water. According to the salinity hazard classes introduced by the United States Lab Classification of Irrigation water [41], 67% of groundwater contains unacceptable salinity levels (C3 and C4 category).

Fluoride concentration ranges from 0.01 to 8.69 mg/L (SD ± 1.65) with a mean value of 1.6 mg/L across the basin (Table 1). Out of 128 samples, 53 samples (41.4%) belong to the category of F^−^ content > 1.5 mg/L. A maximum of 8.69 mg/L was recorded with an average of 3.01 mg/L, indicating that 41.4% of the recorded samples contain F^−^ concentrations more than 2 times the WHO recommended limit. Furthermore, 42% of shallow groundwater samples consist of high fluoride groundwater which ranges from 1.6 to 7.0 mg/L with an average concentration of 3.1 mg/L. High fluoride concentrations have consisted of 40.4% of deep groundwater that ranges from 1.5 to 8.69 mg/L with an average concentration of 2.8 mg/L. Many measured parameters may have a significant contribution to the high fluoride concentration. EC and TDS concentrations of high fluoride groundwater are 2 times more than those of low fluoride groundwater, and all the anions (HCO_3_^−^, SO_4_^2−^, NO_3_^−^, and Cl^−^) have an obvious elevated concentrations than those of low fluoride groundwater.

The existence of high concentrations of NO_3_^−^ in shallow groundwater might be due to the usage of nitrogenous fertilizers, the application of manure, waste disposal, and other human activities and it may affect TDS concentration in shallow groundwater. Higher concentrations of trace metals, Fe, Mn, and Pb have occurred in high fluoride groundwater. According to the Durov diagram (Figure 2), the majority of high TDS groundwater samples are the HCO_3_-Na type, and most samples of high fluoride groundwater are HCO_3_-Na. The groundwater with low fluorine is mainly HCO_3_-Ca, whereas the groundwater with high fluorine is characterized by low Ca^2+^, high Na^+^, and high HCO_3_^−^.

### 4.2. Spatial Distribution of TDS and Fluorides

#### 4.2.1. Horizontal Distribution

The spatial distribution of fluoride and TDS concentrations were obtained using ArcGIS 10.3.1. Figure 3a shows the magnitudes of TDS concentrations. It elaborates into six categories according to the classification of organoleptic properties. The maximum TDS concentration of 10,661 mg/L (DS70) was determined adjacent to the Huangshui River. The central part of the basin has elevated TDS values. TDS concentration tends to increase along the flow paths from the basin margin to the central part. The extremely high TDS concentrated (>1200 mg/L) groundwater is mainly located close to the Huangsui River.

The magnitudes of fluoride concentration are elaborated in five categories as, below 1.5 were to identify areas within WHO permissible limit; 1.5–3.0, above the limit; 3.0–5.0, high concentrations of fluoride; >5.0, extreme concentrations (Figure 3b). The majority of the groundwater samples had elevated fluoride concentrations (>1.5 mg/L) and scattered along the Sanggan River, whereas lesser contents were found along the Huangshui River. The extreme concentrations are mainly in basin center areas due to anthropogenic actions such as agricultural activities. The fluoride concentrations at the basin margin were relatively lower than the central parts. Therefore, both TDS and fluoride concentrations in groundwater showed an elevated trend towards the central part of the basin.

#### 4.2.2. Vertical Distribution

Both shallow and deep groundwater does not show any significant trends of TDS concentration. The high concentrations of TDS mainly occurred in very low depths, ranging from 5 to 40 m (Figure 4a). Only a few samples were recorded as high TDS concentrations in deep aquifers. The shallow groundwater was heavily affected by evaporation and evapotranspiration, which can be the main origin of high TDS groundwater in shallow depths.

Groundwater fluoride concentration has an apparent variation with the well-depth (Figure 4b). The samples were collected from different depths to investigate the drastic changes in fluoride concentration. Approximately 47% of shallow and 39% of deep groundwater samples had fluoride concentrations above the WHO permissible value. The maximum fluoride concentration of shallow groundwater was 7.00 (DS18) and detected at the depth of 48 m, whereas deep groundwater had a maximum of 8.69 mg/L (DD46) at the depth of 55 m. Although many high fluoride concentrations were detected in shallow groundwater, the majority of extreme values were observed in deep groundwater. The genesis of high fluoride concentrations may occur from the ionic activity in deep groundwater that facilitates water-rock reactions and residence time other than the shallow groundwater. However, the majority of the shallow groundwater contained high fluoride concentrations suggesting intensive anthropogenic influences other than natural processes. It is also suspected that the weathering of soil materials may be contributed to fluorides to a great extent within the area [42].

### 4.3. Hydrochemical Evolution Processes

According to the Gibbs anion plot (Figure 5a), the major geochemical process is rock weathering, which dominates TDS concentration in the groundwater. Evapotranspiration is another leading geochemical process that increases TDS concentration, and has a greater influence on the hydrochemical compositions of shallow groundwater (Figure 5a). The relationship between γ(Ca^2+^/Na^+^) and γ(HCO^3−^/Na^+^) can be used to determine the source of major ions in groundwater [5]. The relationship between the ratios of γ(Ca^2+^/Na^+^) and γ(HCO_3_^−^/Na^+^) shows that the groundwater samples are mainly distributed near the end members of silicate salt rocks and close to the end members of evaporites (Figure 5b), indicating that the hydrochemical compositions of groundwater mainly originated from the weathering hydrolysis of silicate minerals and dissolution of evaporites, and relatively less affected by the weathering of carbonate rocks in Datong Basin. High fluorine groundwater is mainly affected by evaporation and karst decomposition.

Hydrolysis is one of the dominant weathering processes that affect the hydrochemistry within the study area. During the hydrolysis process of rock-forming, minerals such as biotite, fluorite, and fluorapatite release Na^+^, K^+^, Ca^2+^, F^−^, etc. Sedimentary aquifers have caused elevated TDS and fluoride concentrations. Na^+^ and K^+^ ions are usually common in most of the rocks that originate from weathering of feldspars (albite and K-feldspar) and micas [40]. A strong positive linear association between Na^+^ and Cl^−^ ion concentrations (Figure 6a) indicates the dissolution of halite may be a major source of Na^+^ and Cl^−^ ions in the solution [43]. The majority of samples are plotted above the 1:1 line, which indicates that excess Na^+^ is remaining in groundwater compared to Cl^−^ ions (Figure 6a). Elevated Na^+^ concentrations can be attributed to the hydrosis of silicate minerals containing sodium and cation exchange process with Ca^2+^ or Mg^2+^ [43,44].

The plot of (Ca^2+^ + Mg^2+^ − HCO_3_^−^ − SO_4_^2−^) versus (Na^+^ − Cl^−^) demonstrates the involvement of Na^+^, Ca^2+^, and Mg^2+^ in the ion exchange reaction (Figure 6b) [45]. The excess Na^+^ and the low Ca^2+^ levels are speculated to be involved in the alternating cation adsorption, and Ca^2+^ in groundwater displaces Na^+^ in clay (Equation (1)).
Ca^2+^ + 2Na-Clay→ 2Na^+^ + Ca-Clay,(1)

### 4.4. Source Appointment by PCA

Principal Component Analysis (PCA) produced the associations among the hydrochemical variables for the identification of underlying hydrogeochemical processes that control groundwater quality, fluoride concentration, and salinity in the study area. The Kaiser’s measure of sampling adequacy (KMO) should be 0.7–0.8 to conduct a factor analysis. Some less-correlated variables were removed using the anti-image correlation [46]. The commonly used Varimax rotation was used to produce a set of uncorrelated variables. The total variance of 89.9% has been discussed under four principal components (PCs) in the PCA analysis (Table 2). The factor loadings greater than 0.5 were considered statistically significant outputs [47]. Standardization is important since high variances of some parameters can be affected by the analyses [48,49]. The data were standardized to their Z-score values to achieve dataset homogeneity and normal distribution.

According to PCA results (Table 2), PC1 was highly weighted by TDS, where TDS concentration exhibits a significant positive loading of 0.97. It is the highest loading obtained from PCA analysis. This component explains 38.9% of the total variance (89.9%) influenced by strong positive loadings of EC, Cl^−^, Mg^2+^, Na^+^, SO_4_^2−^, Ca^2+^, and poor loadings exist for HCO_3_^−^ (positive) and pH (negative). All the anions and cations are significantly associated with TDS in groundwater. The loadings reflect that salinity may be crucial with the prevailing semi-arid climate and weathering of carbonate rocks (calcite and dolomite), and evaporates (sulfates and halides) in the study area. Moreover, the dissolution of minerals is one of the dominant processes that consequently increase the TDS concentration in the groundwater. The component PC2 was highly weighted by fluoride, where F^−^ concentration exhibits a significant positive loading of 0.78. Moreover, this component explains 16.72% of the total variance (89.9%) influenced by strong positive loadings for F^−^, HCO_3_^−^, and pH, and weak negative loadings for Ca^2+^ (Table 2). The pH and F^−^ concentration at the same component reveal that the alkaline pH condition exists in the study area and may be responsible for the fluoride concentration in the solution. Positive loading of F^−^ with poor negative loading of Ca^2+^ may suggest the dissolution and ion-exchange process of fluorine-bearing minerals such as CaF_2_ that release F^−^ ions to the solution of groundwater.

### 4.5. Key Factors Responsible for High TDS and Fluoride Level

TDS and fluoride concentrations in groundwater depend on many natural factors as well as anthropogenic involvements. The natural factors include climate, geogenic formations, residence time, and geochemical interactions. The anthropogenic interventions include agricultural return flows, fertilizer usage, industrial and domestic waste, etc. [1,23,50]. The semi-arid climatic condition has highly a strong influence on the hydrochemistry of shallow groundwater than that of deep groundwater due to evapotranspiration, which is considered one of the major processes that generate high fluoride concentrated saline groundwater.

#### 4.5.1. Evaporation

Isotopic fractionation is an important behavior of water that is used to assess various hydrogeochemical processes within a catchment basin [51]. According to the summary statistics from 42 analyzed samples, stable δ^18^O values range from −12.77 to −7.93 with an average value of −10.62 (SD = 1.48), and the stable δD values range from −97.60 to −61.50 with an average value of −80.15 (SD = 10.45). The results indicate that the groundwater is isotopically fractionated by evapotranspiration and water-rock reactions exist in both shallow and deep groundwater. Shallow groundwater has high δ^18^O values indicating the effect of excessive evapotranspiration in the study area. The prevailing semi-arid climate highly affects the shallow depths compared to the deep depths at Datong. However, high δ^18^O values have occurred in some of the deep groundwater aquifers may be attributed to a longer retention time. Figure 7a reflects the linear regression line equations for both shallow and deep groundwater samples showing good positive linear relationships between the two variables. The fitted lines have a lower slope than the meteoritic lines of the global meteoric water line (GMWL: δD = 8δ^18^O + 10) and Taiyuan Meteoric Water Line (TMWL: δD = 6.66δ^18^O − 3.8), indicating evapotranspiration plays an important effect on both shallow and deep groundwater in the study area.

In Figure 7a, the slope of the “evaporation line” fitted by the hydrogen and oxygen isotope relationship in the study area is smaller than GWML and TWML, suggesting the influence of evaporation. At the same time, high fluoride groundwater is found to be is associated with those samples containing elevated TDS values (Figure 2), indicating the positive correlation of groundwater fluoride to salinity. Those relationships mentioned above proved that evaporation is one of the key factors responsible for elevated TDS in groundwater. The majority of high F^−^ groundwater samples are concentrated in the relatively small F^−^/Cl^−^ ratio in the range of 0~0.2 (Figure 8a). The relation shows that evaporation is one of the factors affecting F^−^ enrichment in groundwater.

The relationship between Cl^−^ and δ^18^O can indicate the fractionation process of various isotopes during groundwater transport [52]. There are three trends in the relationship between Cl^−^ concentration and δ^18^O in groundwater in the study area (Figure 7b): (1) The δ^18^O increased sharply, whereas the Cl^−^ concentration hardly changed. This reflects fast recharge by irrigation returns and mixing with shallow water because both are associated with different δ^18^O values.; (2) There is a significant correlation between δ^18^O and Cl^−^ concentration, and δ^18^O tends to enrich with the increase in Cl^−^ concentration, indicating that the groundwater is affected by evaporation. (3) The trend slope of Cl^−^ is smaller than that of evaporation, and the concentration of Cl^−^ is high, which should be resulted from halite dissolution. Part of the samples in Figure 8b are located around trend line 2, confirming the influence of evaporation, but most of the samples are located near trend line 1 and trend line 3, indicating that evaporation is just one of the factors affecting solute transport in the saquifers, and it is also affected by mixing and leaching.

#### 4.5.2. pH and Alkalinity

pH is an important parameter that affects F^−^ adsorption on mineral surfaces [53]. The pH of the majority of elevated F^−^ groundwater ranges from 7.5 to 8.5 (Figure 8b). Weak alkaline condition is favorable for fluoride enrichment that assists in OH^−^−F^−^ exchange reactions in the aqueous phase [23,35,43,54]. Fluoride ion has the same charges with a similar radius as OH^−^ ion and, can exchange each other in mineral structures. Therefore, F^−^ ions adsorped by the clay minerals in the sedimentary aquifers can be displaced by OH^−^ ions at high pH, and release to the groundwater [23,54]. Furthermore, groundwater F^−^ content shows a moderately strong correlation with alkalinity (Figure 8d). The elevated alkalinity in groundwater facilitates the mobilization of F^−^ from weathered rocks [55]. The moderate alkalinity prevailing in clay and sandy loam deposits facilitates s F^−^ release to the groundwater from sedimentary aquifers.

TDS concentrations show a decreasing trend with the increase in pH (Figure 8c). The majority of high TDS concentrations occurred at the pH ranging from 7.2- 7.8, when relatively weak alkaline conditions exist. Therefore, weakly alkaline conditions may be favorable for the formation of high TDS concentration within the area. The HCO_3_^−^ion is the predominant ion in both low F- and high F- groundwater, which is considered a dominant hydrochemical component leading to high F- groundwater [56]. HCO_3_^−^ concentration shows a strong positive (r = 0.66) relationship with F^−^ content (Figure 8c). It is noticed that bicarbonates may be mobilized to the groundwater through leaching and dissolution processes of fluoride-bearing minerals [23,42,57]. The dissolution process of fluorite with elevated HCO_3_^−^ concentrations is thermodynamically convenient [57]. The dissolution reaction of fluorite is shown in Equation (2).
CaF_2_ + 2HCO_3_^−^→ 2F^−^ + CaCO_3_ + H_2_O + CO_2_(2)

In Figure 8d, groundwater HCO_3_^−^ concentrations show a moderate positive (r = 0.36) relationship with F^−^ content, indicating competitive adsorption between F^−^ and HCO_3_^−^ that influences F^−^ enrichment in groundwater [24] (Liu et al., 2015).

#### 4.5.3. Cation Exchange

Sodium is one of the major cations in both low and high F^−^ groundwater that favors F^−^ enrichment process [58]. The majority of groundwater samples were identified as the HCO_3_-Na type water, which has a possibility of ion exchange between Na^+^ and Ca^2+^ ions under alkaline pH, enhancing F^−^ leaching into the groundwater. Figure 6b and Figure 8e demonstrate that the F- enrichment mechanism could be related to the cation exchange process. Previous investigations have revealed that cation exchange is one of the geochemical processes that significantly influence the groundwater chemistry at the Datong Basin [23,59,60,61]. This cation exchange process accelerates fluorite dissolution by removing Ca^2+^ from the solution, elevating F^−^ concentration in groundwater.

Furthermore, it is considered one of the main contributing factors to groundwater salinity prevailing within the area. Two indexes, CAI-1 and CAI-2, were used to illustrate the possibility of cation exchange, and the calculation methods were shown in Equations (3) and (4), respectively.
(3)$$\mathrm{CAI-1}=\frac{{\mathrm{Cl}}^{-}-\left({\mathrm{Na}}^{+}+K^{+}\right)}{{\mathrm{Cl}}^{-}}$$
(4)$$\mathrm{CAI-2}=\frac{{\mathrm{Cl}}^{-}-\left({\mathrm{Na}}^{+}+K^{+}\right)}{{\mathrm{HCO}}_{3}^{-}+{\mathrm{SO}}_{4}^{2-}+{\mathrm{CO}}_{3}^{2-}+{\mathrm{NO}}_{3}^{-}}$$

When the ions of Na^+^ and K^+^ are replaced by Ca^2+^ and Mg^2+^ in water, the values of CAI-1 and CAI-2 will be positive. On the contrary, when Ca^2+^ and Mg^2+^ in water exchange Na^+^ and K^+^ in adsorbed states, the values of CAI-1 and CAI-2 will be negative. It can be seen from Figure 8f, these two indexes are negative for most water samples, which also confirms that cation exchange occurs in high F^−^ groundwater, and mainly the Ca^2+^ and Mg^2+^ exchange of Na^+^ and K^+^ in adsorption states. Therefore, the reduction in Ca^2+^ concentration in groundwater caused by cation exchange promotes the enrichment of F^−^ in groundwater.

The majority of high F^−^ concentrations occurred under low concentrations of Ca^2+^ (Figure 9a), indicating that Ca^2+^ could inhibit F^−^. Because Ca^2+^ has a strong affinity with HCO_3_^−^ and CaCO_3_ precipitates, which reduces Ca^2+^ levels in groundwater and accelerates the dissolution of fluorite (Equation (5)), thus increasing the concentration of F^−^ in groundwater.
CaF_2_ + 2HCO_3_^−^ = CaCO_3_ + 2F^−^ + H_2_O + CO_2_(5)

It can be seen from Figure 9b that the fluorite in the groundwater is unsaturated, whereas the calcite is oversaturated in the study area. The fluorite saturation index is positively correlated with F^−^, indicating that the F^−^ concentration in the groundwater in the study area is mainly derived from the weathering dissolution of fluorite. Ca^2+^ ions may enter the solution from other minerals such as calcite and feldspar, which can reduce and facilitate fluorite dissolution via precipitating as calcite [33]. It can be suggested that the solubility of fluorite and calcite might dominate the hydrochemical compositions of groundwater including, F^−^, Ca^2+^, and HCO_3_^−^ concentrations [57].

When the activity of Ca^2+^ and F^−^ in solution is greater than the fluorite dissolution equilibrium constant, the dissolution equilibrium of fluorite will move towards precipitation [45]. Samples are concentrated below the fluorite dissolution equilibrium line (pK_fluorite_ = 10.6) (Figure 10), indicating that the F^−^ concentration in groundwater samples is controlled by the fluorite solubility in the study area. When only fluorite was dissolved, the activity of F^−^ and Ca^2+^ increased along trend line 1; however, most of the samples were located to the right of trend line 1, indicating that Ca^2+^ in the groundwater had other sources besides fluorite. Given that groundwater contains large amounts of HCO_3_^−^ and SO_4_^2−^, groundwater Ca^2+^ may come from dissolved calcite, dolomite, and gypsum. When calcite and fluorite were dissolved at a mass ratio of 200:1, the activity of F^−^ and Ca^2+^ increased along trend line 2, and most water samples were located between trend line 1 and trend line 2, indicating that the concentration of F^−^ was controlled by Ca^2+^ from dissolved sources of fluorite, calcite, gypsum and other minerals.

## 5. Conclusions

The study area has a large spatial heterogeneity of groundwater chemistry. Fluoride and TDS concentrations have an increasing trend towards the central part of the basin. 32.8% of groundwater samples contained elevated TDS concentrations, which exceed the WHO drinking water guideline value of 1000 mg/L. In total, 41.4% of groundwater samples had F^−^ concentrations exceeding 1.5 mg/L recommended by the WHO guideline for drinking water quality. The interaction between groundwater and aquifer sediments as well as evaporation jointly controls the enrichment of F^−^ and TDS in groundwater. The dominant geochemical processes at Datong Basin include the hydrolysis of silicate, and the dissolution–precipitation of carbonates, as well as evaporates and cation exchange. Dissolution and precipitation of F-bearing minerals and carbonate are the dominant processes dominating groundwater F^−^ levels. Alkaline pH conditions, moderate TDS and Na^+^, high HCO_3_^−^, and lower Ca^2+^ concentrations facilitate the enrichment of fluoride in the study area. The evapotranspiration process can be also the most influential factor responsible for high F^−^ and TDS content, because of the extended residence time of groundwater and low rainfall in the central part of the Datong Basin.

## Figures and Tables

**Figure 1 ijerph-20-01832-f001:**
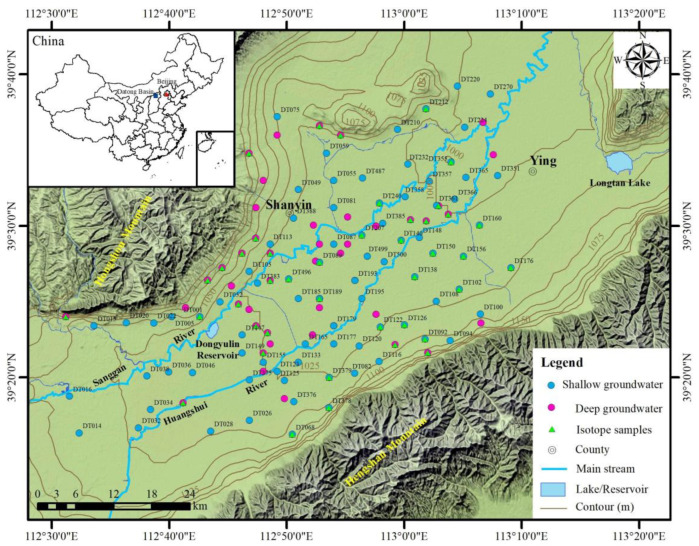
Topography and sampling sites of the study area.

**Figure 2 ijerph-20-01832-f002:**
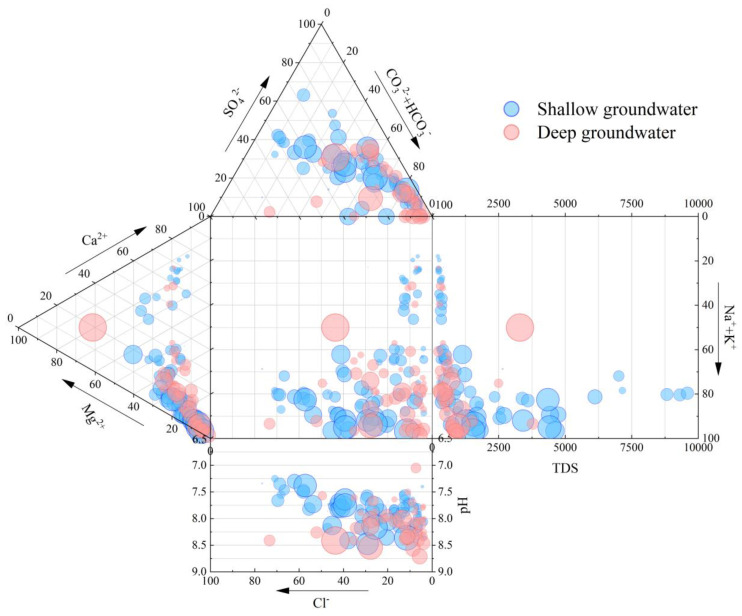
Durov diagram with fluoride concentrations indicated as bubbles.

**Figure 3 ijerph-20-01832-f003:**
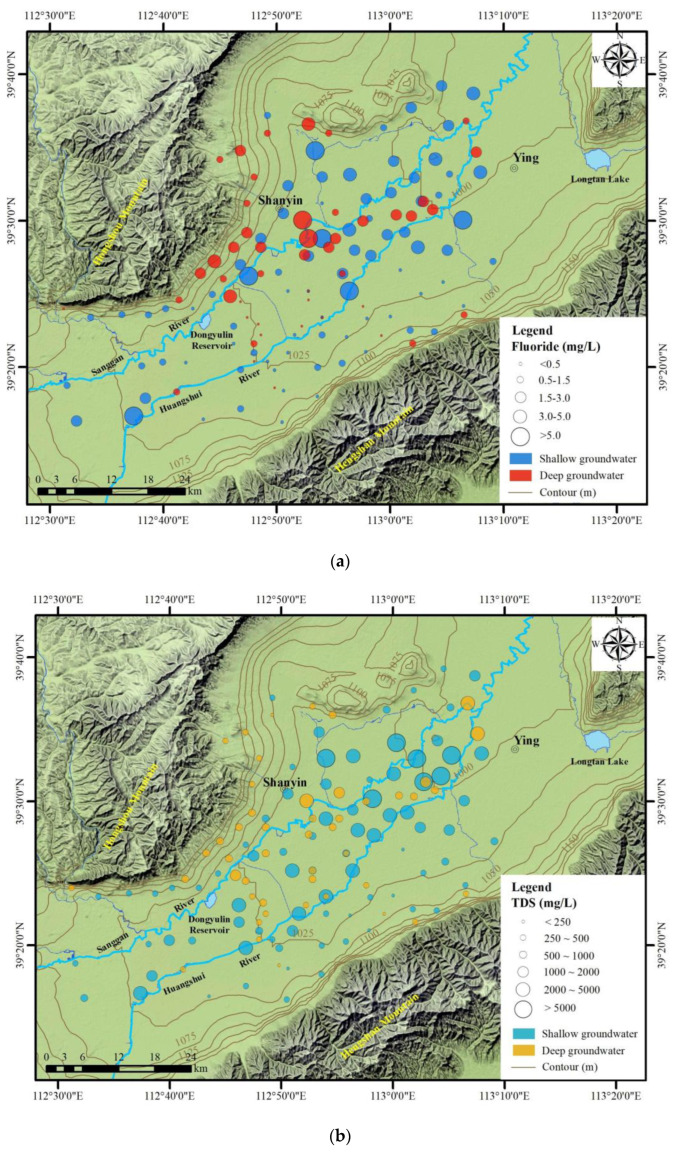
Horizontal distribution of groundwater (**a**) fluoride and (**b**) TDS concentrations in the study area.

**Figure 4 ijerph-20-01832-f004:**
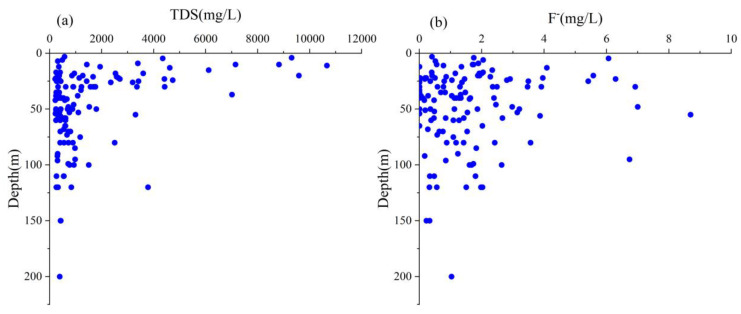
Vertical distribution of groundwater (**a**). TDS and (**b**) Fluoride concentrations in the study area.

**Figure 5 ijerph-20-01832-f005:**
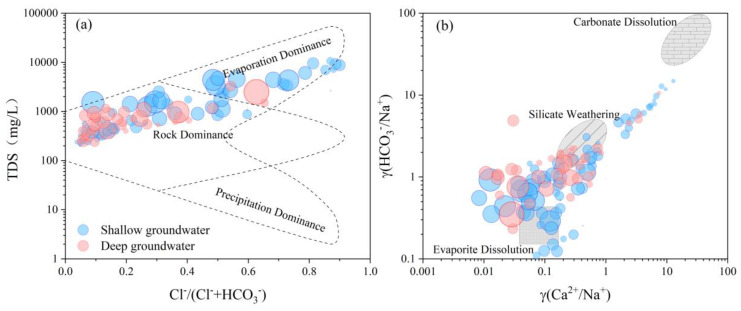
(**a**) Na^+^/(Na^+^ + Ca^2+^) mg/L vs. Log TDS with fluoride concentrations indicated as bubbles; (**b**) Na-normalized HCO_3_^−^ vs. Na-normalized Ca^2+^ (mM/mM) with fluoride concentrations indicated as bubbles.

**Figure 6 ijerph-20-01832-f006:**
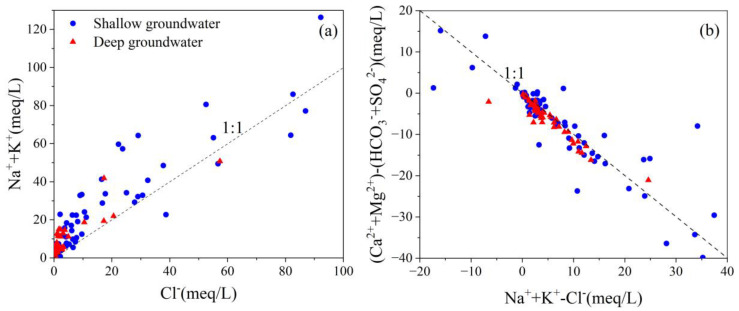
(**a**) (Na^+^ + Ca^2+^) mg/L vs. Cl^−^ mg/L; (**b**) (Ca^2+^ + Mg^2+^)-(SO_4_^2−^ − HCO_3_^−^)meq/L vs. (Na^+^ + K^+^ − Cl^−^) meq/L.

**Figure 7 ijerph-20-01832-f007:**
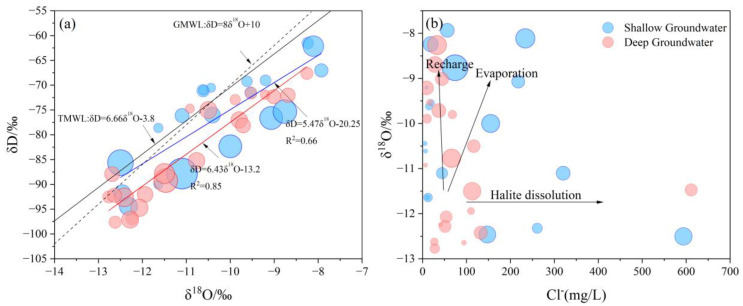
(**a**) Bivariate plot of δD and δ^18^O in groundwater samples with TDS concentrations indicated as bubbles; (**b**) Bivariate plot of Cl and δ^18^O in groundwater samples with fluoride concentrations indicated as bubbles.

**Figure 8 ijerph-20-01832-f008:**
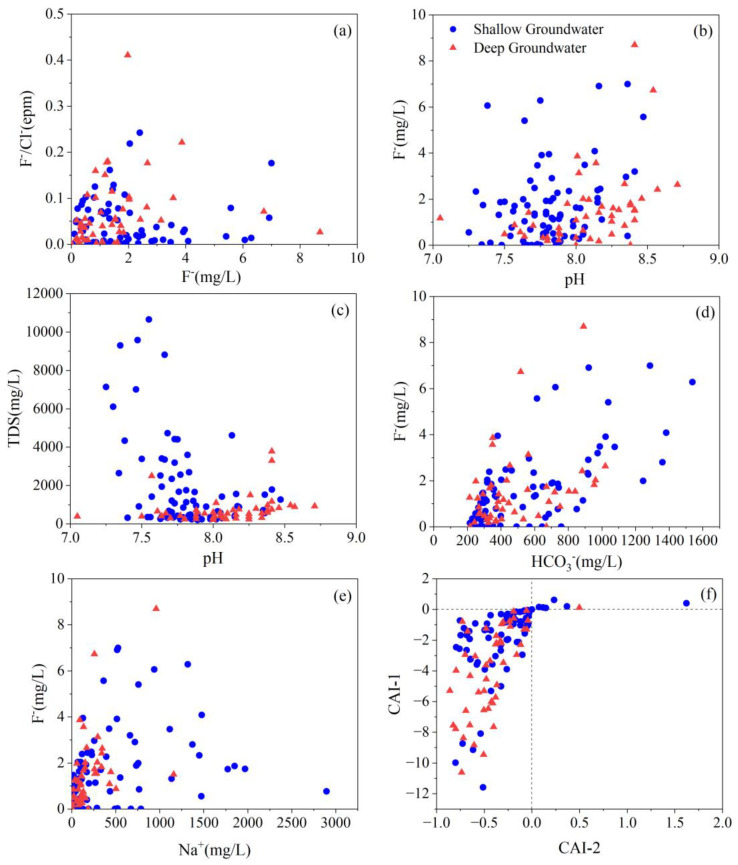
Scatter plots of (**a**). F^−^ vs.F^−^/Cl^−^ (epm); (**b**). pH vs. F^−^; (**c**). pH vs. TDS; (**d**). HCO_3_^−^ vs. F^−^; (**e**). Na^+^ vs. F^−^; (**f**). CAI-2 vs. CAI-1^−^.

**Figure 9 ijerph-20-01832-f009:**
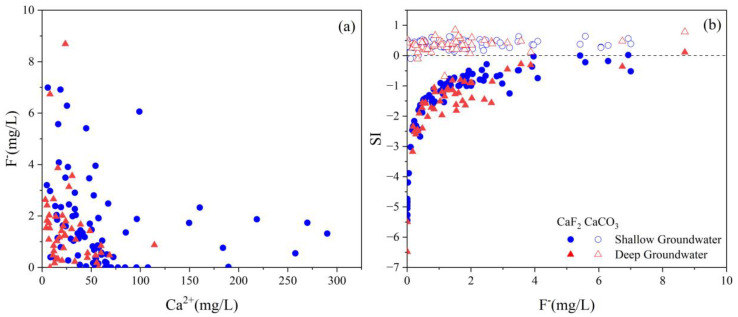
(**a**) Scatter plots of Ca^2+^ vs. F^−^; (**b**) SI of calcite and fluorite in groundwater.

**Figure 10 ijerph-20-01832-f010:**
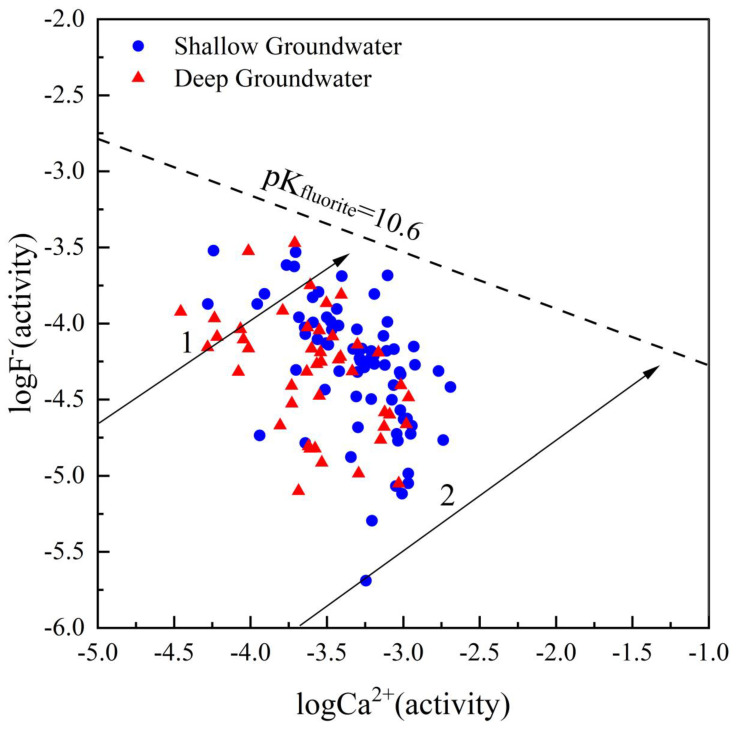
logF^−^ (activity) vs. logCa^2+^ (activity).

**Table 1 ijerph-20-01832-t001:** Statistical results of the hydrochemical analysis of groundwater.

Chemical Parameters	Min	Max	Mean	SD	China- Standard for Groundwater Quality (GB/T 14848-2017) [36]	WHO Guideline for Drinking Water Quality (2017) [35]
T (°C)	9.6	17.8	12.1	1.6	-	-
pH	7.05	8.71	7.91	0.31	-	-
EC (µs/cm)	206	22,600	2531	3581	-	-
TDS (mg/L)	208.9	10,661	1462	2021	1000	1000
F^−^ (mg/L)	0.01	8.69	1.61	1.65	1	1.5
Cl^−^ (mg/L)	5.31	3272	336.7	650.8	250	250
NO_3_^−^ (mg/L)	0.05	1118	63.41	160.2	20	50
SO_4_^2−^(mg/L)	0.05	4456	317.9	630.8	250	250
HCO_3_^−^ (mg/L)	214	1537	525.9	288.2	-	-
K^+^ (mg/L)	0.01	326.8	7.99	31.21	-	-
Na^+^ (mg/L)	5.89	2895	342.3	479.3	200	-
Ca^2+^ (mg/L)	3.22	290	48.68	50.85	-	-
Mg^2+^ (mg/L)	4.30	773.3	80.1	134	-	-
Li (mg/L)	0.01	0.33	0.07	0.05	-	-
Ba (mg/L)	0.06	2.23	0.64	0.399	0.7	0.7
Sr (mg/L)	0.17	8.07	1.25	1.43	-	-
Fe (mg/L)	0.00	0.71	0.03	0.08	0.3	Not exceeding 0.1
Mn (mg/L)	0.00	1.12	0.08	0.17	0.1	Not exceeding 0.05

**Table 2 ijerph-20-01832-t002:** Rotated Component Matrix of PCA Analysis.

	Component			
	PC1	PC2	PC3	PC4
TDS	0.97	0.06	0.21	0.1
EC	0.95	0.04	0.22	0.07
Cl^−^	0.94	−0.05	0.16	0.06
Mg^2+^	0.94	−0.15	0.17	0.02
Na^+^	0.94	0.23	0.14	0.11
SO_4_^2−^	0.93	−0.03	0.13	0.02
Ca^2+^	0.72	−0.53	0.34	0.02
F^−^	0.09	0.78	0.47	0.08
HCO_3_^−^	0.49	0.72	−0.05	0.24
pH	−0.37	0.69	−0.18	−0.22
NO_3_^−^	0.46	0.03	0.79	0.24
K^+^	0.03	0.03	0.15	0.97
Eigenvalues	7.07	2.01	1.14	0.57
Variance (%)	58.93	16.72	9.47	4.77
Cumulative (%)	58.93	75.66	85.13	89.9

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
