# Peer review of "Origin and Enrichment Mechanisms of Salinity and Fluoride in Sedimentary Aquifers of Datong Basin, Northern China"

_ijerph, 2023, doi:10.3390/ijerph20031832_

Round 1
Reviewer 1 Report
I would like to emphasize a few significant facts. First of all, why didn't you use Kriging for display of geospatial distribution of groundwater? Kriging is namely a group of geostatistical methods mainly used for this purpose - why did you decide for IDW interpolation method? Secondly, you examined the quality of water in order to use it for drinking, but there is no any information about this in conclusion? You have to summarize if it can be used as drinking water without any processing. Your discussion also needs to be a bit extended - you mainly took references related to standards and guidelines, but just some of them are connected with some papers. Try to find some papers that deal with water quality and put it into Introduction. I am giving you one that you should add: Nikolić, V., Jokanović, D., Petrović, J., AnÄ‘elković, A. The assessment of water quality in hygrophilous forests of Ravni Srem. Šumarstvo 2018, 1-2: 155-166. In the introductory part you should add this reference: There are a lot of papers dealing with water quality from all over the world (Nikolić et al., 2018;.....). All in all, the paper is of a very good quality and demands just these minor revisions.
Reviewer 2 Report
Dear Editor
Comments on manuscript titled: “Origin and enrichment mechanisms of salinity and fluoride in sedimentary aquifers of Datong Basin, Northern China”
Abstract & Introduction
1. Add innovation of the research to Introduction Section.
2. See the previous researches as follows:
ü Establishing a Data Fusion Water Resources Risk Map Based on Aggregating Drinking Water Quality and Human Health Risk Indices
ü Using a soft computing OSPRC risk framework to analyze multiple contaminants from multiple sources; a case study from Khoy Plain, NW Iran
ü Qualitative risk aggregation problems for the safety of multiple aquifers exposed to nitrate, fluoride and arsenic contaminants by a ‘Total Information Management’framework
ü Supervised committee machine with artificial intelligence for prediction of fluoride concentration
ü Hydrogeochemical Analysis for Tasuj Plain Aquifer, Iran
3. Add a Table to compare previous researches.
Methology and Study area:
4. Line 98: The potential evaporation is correct
5. Give more information about Geology and Hydrogeology of study area
6. Give Geology map of study area
Result and Discussion
7. Line 206: this sentence is not true, please check
8. Figure 2: Please add expansive Durove diagram, If it is possible
9. Give origin, source and pass way of contamination
10. Give groundwater direction in study area
11. Line 327: what is the source of fluoride? Evaporate rock or CaF2
12. Figure 8b show low correlation between, please discuss
Reviewer 3 Report
In line 313 total variance is 81,25%, and in line 322 is 89,9% as in Table 2. Why is that?
in line 366 to be is associated is has to be deleted
in line 391 Fig 7b is Figure 7c
Round 2
Reviewer 2 Report
The comments are addressed approperiately